# Spectrin regulates Hippo signaling by modulating cortical actomyosin activity

Hua Deng, Wei Wang, Jianzhong Yu, Yonggang Zheng, Yun Qing, Duojia Pan*

Department of Molecular Biology and Genetics, Howard Hughes Medical Institute, Johns Hopkins University School of Medicine, Baltimore, United States

**Abstract** The Hippo pathway controls tissue growth through a core kinase cascade that impinges on the transcription of growth-regulatory genes. Understanding how this pathway is regulated in development remains a major challenge. Recent studies suggested that Hippo signaling can be modulated by cytoskeletal tension through a Rok-myosin II pathway. How cytoskeletal tension is regulated or its relationship to the other known upstream regulators of the Hippo pathway remains poorly defined. In this study, we identify spectrin, a contractile protein at the cytoskeleton-membrane interface, as an upstream regulator of the Hippo signaling pathway. We show that, in contrast to canonical upstream regulators such as Crumbs, Kibra, Expanded, and Merlin, spectrin regulates Hippo signaling in a distinct way by modulating cortical actomyosin activity through non-muscle myosin II. These results uncover an essential mediator of Hippo signaling by cytoskeleton tension, providing a new entry point to dissecting how mechanical signals regulate Hippo signaling in living tissues.

## Introduction

The Hippo signaling pathway controls organ size in *Drosophila* through coordinated regulation of cell growth, proliferation, and apoptosis (*Harvey and Tapon, 2007*; *Pan, 2007*; *Halder and Johnson, 2011*). This pathway involves a core kinase cascade in which the Hippo-Salvador (Hpo-Sav) kinase complex phosphorylates and activates the Warts-Mats (Wts-Mats) kinase complex, which in turn inactivates the Yorkie (Yki) oncoprotein through phosphorylation. This phosphorylation event excludes Yki from the nucleus, where it normally functions as a coactivator for the expression of Hippo pathway target genes. The conserved function of Hippo signaling in mammalian growth control and tumorigenesis has stimulated much interest in understanding the regulation of this pathway in development, regeneration, and disease (*Zhao et al., 2008*; *Pan, 2010*; *Barry and Camargo, 2013*; *Harvey et al., 2013*; *Johnson and Halder, 2014*).

Genetic studies in *Drosophila* suggest that the Hippo kinase cascade is modulated by a diverse array of upstream regulators (*Boggiano and Fehon, 2012*; *Enderle and McNeill, 2013*). Prominent among these are three membrane-associated tumor suppressor proteins, Expanded (Ex), Merlin (Mer) and Kibra, which act semi-redundantly to activate downstream signaling, by recruiting the core kinase cassette to the plasma membrane or cytoplasmic sequestration of Yki through direct binding. Other tumor suppressor proteins implicated as upstream regulators of the Hippo kinase cascade include the atypical cadherins Fat (Ft) and Dachsous (Ds), apical basal polarity regulators Crumbs (Crb), Scribble (Scrib), Discs large (Dlg), and Lethal giant larvae (Lgl), the Ste20-like kinase Tao-1, the protein tyrosine phosphatase Pez, and the cell adhesion molecule Echinoid (Ed). At least some of these tumor suppressors have been shown to play a conserved role in Hippo signaling in mammals, which have also acquired additional regulators such as Angiomotin (Amot), α-catenin, and G protein-coupled receptors (GPCRs) (*Yu and Guan, 2013*).

In an exciting recent development, studies in cultured mammalian cells have implicated YAP and TAZ, the mammalian counterpart of Yki, as key mediators of mechanotransduction, whereby changes

*For correspondence: djpan@jhmi.edu

**eLife digest** Organs including the liver, eyes, and lungs are made up of millions of cells, and how these organs stop growing once they reach their final size has fascinated scientists for decades. The cells in developing organs must communicate with each other and respond appropriately to the signals that they receive from other cells. This requires so-called "signaling pathways". One such pathway that involves a protein called Hippo is known to control when cells should grow and divide and when they should stop. If this pathway does not work correctly, it can cause too many cells to be formed, which may result in cancer.

The Hippo signaling pathway can also be regulated by an extensive network of protein filaments found within cells, called the cytoskeleton. This network can exert forces on the cells, which can have a major impact on cell growth. However, the mechanism behind the interaction between the cytoskeleton and the Hippo signaling pathway is poorly understood.

Now, Deng et al. have engineered fruit flies in which the expression of individual genes had been artificially reduced, and looked for flies that had enlarged wings. Three genes identified in these experiments encode different subunits of a large spring-like protein, called spectrin, which is part of the cytoskeleton. This suggests that normally spectrin limits wing size. Furthermore, spectrin was also found to control the size of other organs in the fruit flies, such as the eyes and ovaries. In all of these organs, the Hippo signaling pathway failed to work properly in the absence of spectrin. Deng et al. then further explored the relationship between spectrin and Hippo signaling and found that cells without spectrin show abnormally high levels of tension in their cytoskeleton. When flies that lacked spectrin were engineered to reduce this tension, these flies developed normal sized organs. These findings reveal the importance of cytoskeleton tension in controlling tissue growth, and provide a new entry point to understand how normal tissues grow to their characteristic size and how such process goes awry in cancer.

in cell–extracellular matrix (ECM) interaction, cell shape, or the actomyosin cytoskeleton influence cellular behaviors such as proliferation and differentiation (*Dupont et al., 2011*; *Wada et al., 2011*; *Aragona et al., 2013*). Molecular interrogation of this mechanotransduction process suggests that the subcellular localization and thus the activity of YAP/TAZ is regulated by the contractile actomyosin through a Rok (Rho-associated protein kinase)-myosin II pathway. In *Drosophila*, excessive actin polymerization or activation of Rok-myosin II also lead to increased Yki activity (*Fernandez et al., 2011*; *Sansores-Garcia et al., 2011*; *Rauskolb et al., 2014*), suggesting that cytoskeleton tension is a conserved regulator of Hippo signaling in diverse animals. Despite these exciting progresses, important questions remain: what is the relationship between the actomyosin cytoskeleton and the other reported upstream regulators such as Kibra, Ex, and Mer in Hippo pathway regulation? Furthermore, since mechanical force can be sensed and transduced in many subcellular regions such as cell–cell junction, plasma membrane-associated cytoskeleton cortex, cytoplasmic stress fiber and even nuclear membrane-associated cortex, what is the exact nature of the cytoskeletal force that is relevant to Hippo signaling?

Spectrin is a large spring-like protein that forms the spectrin-based membrane skeleton (SBMS) right beneath the plasma membrane by crosslinking short F-actin and binding integral membrane proteins (*Bennett and Baines, 2001*). Spectrin proteins are conserved in all eukaryotes from protozoa to humans (*Baines, 2009*). Spectrin exists mainly as heterotetramers of α and β subunits, in which the α and β subunits are assembled side to side in an antiparallel fashion to form rod-like αβ dimers that in turn self-associate head to head to form tetramers. With the help of Adducin protein, these tetramers crosslink with short F-actin to form a lattice-like network that supports the structural stability of the plasma membrane. Indeed, spectrin was first identified as proteins that contribute to the mechanical resilience of erythrocytes (*Bennett and Gilligan, 1993*), and mutations in spectrin genes result in severe anemia characterized by abnormally shaped erythrocytes with increased membrane fragility (*Delaunay, 2007*). A recent study reported that the SBMS functions independently of actin dynamics to maintain the pre-stress status of touch receptor neurons (TRNs) in *Caenorhabditis elegans* and therefore enhances the overall mechanosensitivity of these neurons (*Krieg et al., 2014*). Whether the SBMS plays a direct role in mechanotransduction, or the relationship between the SBMS and the actomyosin cytoskeleton in mechanotransduction, is less clear.

The fruit fly *Drosophila* encodes one α subunit (α-Spec) and two β subunits (β-Spec and βHeavy-spec or βH-Spec), which generate two spectrin tetramers, $(\alpha\beta)_2$ and $(\alpha\beta H)_2$. In *Drosophila* ovarian follicle cells, β-Spec and βH-Spec are localized to the basolateral and apical membrane, respectively, while α-Spec is localized along the entire apical–basal axis (*Lee et al., 1997*). Here, we report the identification of spectrin genes as negative growth regulators and upstream regulators of the Hippo signaling pathway in *Drosophila*. Interestingly, unlike the previously reported upstream regulators of the Hippo pathway such as Crumbs, Kibra, Ex and Mer, spectrin regulates Hippo signaling through a distinct mechanism by modulating the activity of non-muscle myosin II. These results uncover an essential mediator of Hippo signaling by cytoskeleton tension at the membrane–cytoskeleton interface, providing a new entry point to dissecting how mechanical signals regulate Hippo signaling in living tissues.

## Results

In a genome-wide RNAi screen, we identified *α-spec*, *β-spec*, and *βH-spec* as tumor suppressors based on the enlarged wing phenotype produced by Gal4-mediated overexpression of UAS-RNAi transgenes in the wing tissue (*Figure 1A–B*). Antibody staining confirmed that the RNAi transgenes of *α-spec* and *βH-spec* efficiently knocked down the expression of the respective genes in the imaginal discs (*Figure 1—figure supplement 1*). Furthermore, consistent with previous studies in ovarian follicle cells (*Lee et al., 1997*), βH-Spec is mainly localized to the apical membrane of the imaginal disc epithelial cells (*Figure 1—figure supplement 1E–E''*), while α-Spec localizes to both lateral and apical domains in these cells (*Figure 1—figure supplement 1B–B''*). Given that RNAi knockdown of any of the three spectrin genes produced a similar phenotype and that α-Spec is the major component of both apical and lateral SBMS in the epithelial cells (*Lee et al., 1997*), we focused our analysis on *α-spec* unless otherwise indicated.

The wing overgrowth phenotype resulting from *α-spec* RNAi was suppressed by Wts overexpression (*Figure 1A–B*), suggesting a potential relationship between the SBMS and Hippo signaling. To explore this relationship further, we tested whether spectrin knockdown enhances the mild overgrowth phenotype caused by mutations in upstream regulators of the Hippo signaling, considering that they usually work redundantly to regulate Hippo signaling (*Hamaratoglu et al., 2006*). Reducing *α-spec* expression in *ex^{e1}* mutant cells significantly enhanced the overgrowth phenotype caused by *ex^{e1}* in both the adult notum and eye tissue (*Figure 1C–D*). Another hallmark of defective Hippo signaling is the alteration of interommatidial cell number in pupal retina. Wild-type eyes have an average of 12 interommatidial cells surrounding each unit eye (*Carthew, 2007*) (*Figure 1E*). Mutants of upstream regulators of the Hippo pathway display a mild increase in interommatidial cells, with loss of *ex*, *mer*, and *kibra* resulting in 1.3, 4.2, and 3.8 extra cells per cluster (ECPC), respectively (*Figure 1E–L*). Pupal retina with *α-spec* RNAi had an average of 2.4 ECPC (*Figure 1F*), which is stronger than *ex*, but is milder than *mer* or *kibra* mutants. Interestingly, combinations of *α-spec* RNAi with any of these mutations resulted in a striking synergistic phenotype, with *α-specRNAi;ex*, *α-specRNAi;mer*, and *α-specRNAi;kibra* double mutant retina producing 17.7, 24.5 and 10.3 ECPC, respectively (*Figure 1E–L*). Like *α-spec*, RNAi of *β-spec* or *βH-spec* also led to a mild increase of interommatidial cells (*Figure 1—figure supplement 2A–B''*), which were confirmed in mutant clones for a null allele of *α-spec* (*α-spec^{rg41}*) or *β-spec* (*β-spec^C*) (*Figure 1—figure supplement 2C–D''*). Taken together, these results suggest that spectrin functions in conjunction with the known upstream regulators of the Hippo pathway to regulate tissue growth.

To further corroborate our hypothesis that spectrin is an upstream regulator of the Hippo signaling, we examined the expression of *ex*, a well-characterized Hippo target gene, in imaginal discs. Clones with *α-spec* RNAi showed a significant increase in Ex protein levels and a modest increase in *ex* transcription in the pupal retina (*Figure 2A–C'*). Similarly, *α-spec* RNAi by the engrailed-Gal4 led to a significant increase in Ex protein and transcription levels (*Figure 2—figure supplement 1A–C''*). Consistent with the upregulation of Hippo target genes, increased nuclear accumulation of Yki was observed in *α-spec^{rg41}* mutant cells (*Figure 2D–D'''*). We next examined the expression of *diap1*, another well-characterized Hippo target gene. Similar to *mer* or *kibra* mutant clones, which display no obvious increase in *diap1* expression (*Pellock et al., 2007*; *Yu et al., 2010*), clones with *α-spec* RNAi showed no visible increase in Diap1 protein levels (*Figure 2E–E'*). Interestingly, combinations of *α-spec* RNAi with any of these mutations (*α-spec RNAi;kibra^{del}* or *α-spec RNAi;mer^4*) resulted a significant elevation in Diap1 protein levels (*Figure 2F–G'* and *Figure 2—figure supplement 1D–E''*), further implicating spectrin as an upstream regulator of Hippo signaling.

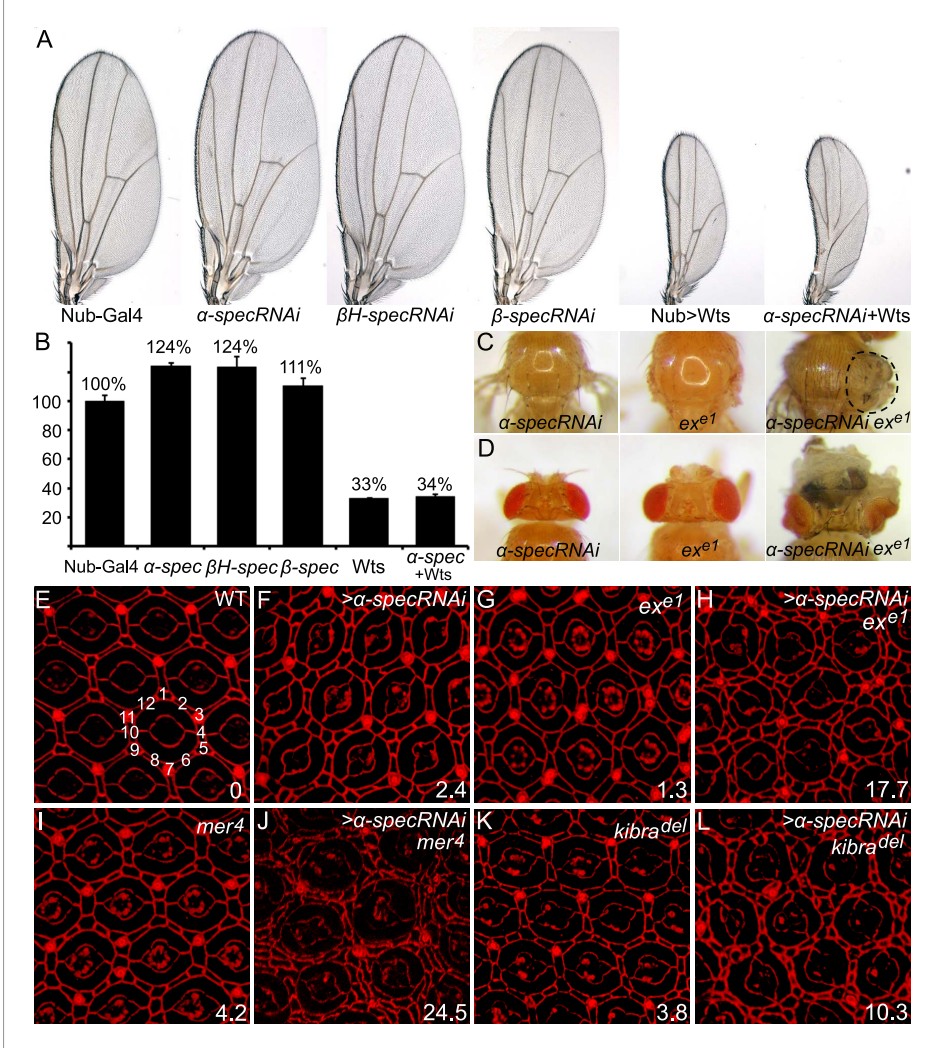

**Figure 1**. Spectrins function synergistically with upstream regulators of Hippo signaling to control tissue growth. (**A**–**B**) RNAi transgene against each of the three *spectrin* genes and *UAS-wts* were expressed separately or in combination in the wing tissue by *nub*-Gal4. Representative adult wings are shown. The graph in **B** shows quantification of wing size relative to nub-Gal4/+ control (mean ± SEM, n = 15). (**C**–**D**) MARCM clones with *α-spec* RNAi, *ex^{e1}* mutation or their combination were produced in the notum (**C**) or in the eye tissues (**D**). Note the massive overgrowth (circled area) only in flies containing *ex* mutant clones with *α-spec* RNAi. (**E**–**L**) Pupal eye discs of the indicated genotypes were stained for DE-cad. Twenty ommatidial clusters of each genotype were used for counting interommatidial cells, and the number on the lower right of each panel indicates the number of extra cells per cluster (ECPC).

The following figure supplements are available for figure 1:

**Figure supplement 1**. Analysis of α-Spec and βH-Spec localization and RNAi knockdown efficiency in imaginal disc epithelial cells.

**Figure supplement 2**. RNAi of *βH-spec* or *β-spec*, or a null allele of *α-spec* or *β-spec*, phenocopies the overgrowth phenotype of *α-spec* RNAi in pupal retina.

---

To test whether spectrin is a regulator of Hippo signaling beyond the imaginal discs, we examined the *Drosophila* ovary, where Hippo signaling is required in the posterior follicle cells (PFCs) for a mitosis-to-endoreplication switch between stages 6 and 7. Complete loss of Hippo signaling, as shown by mutants lacking components of the core kinase cascade, results in prolonged expression of

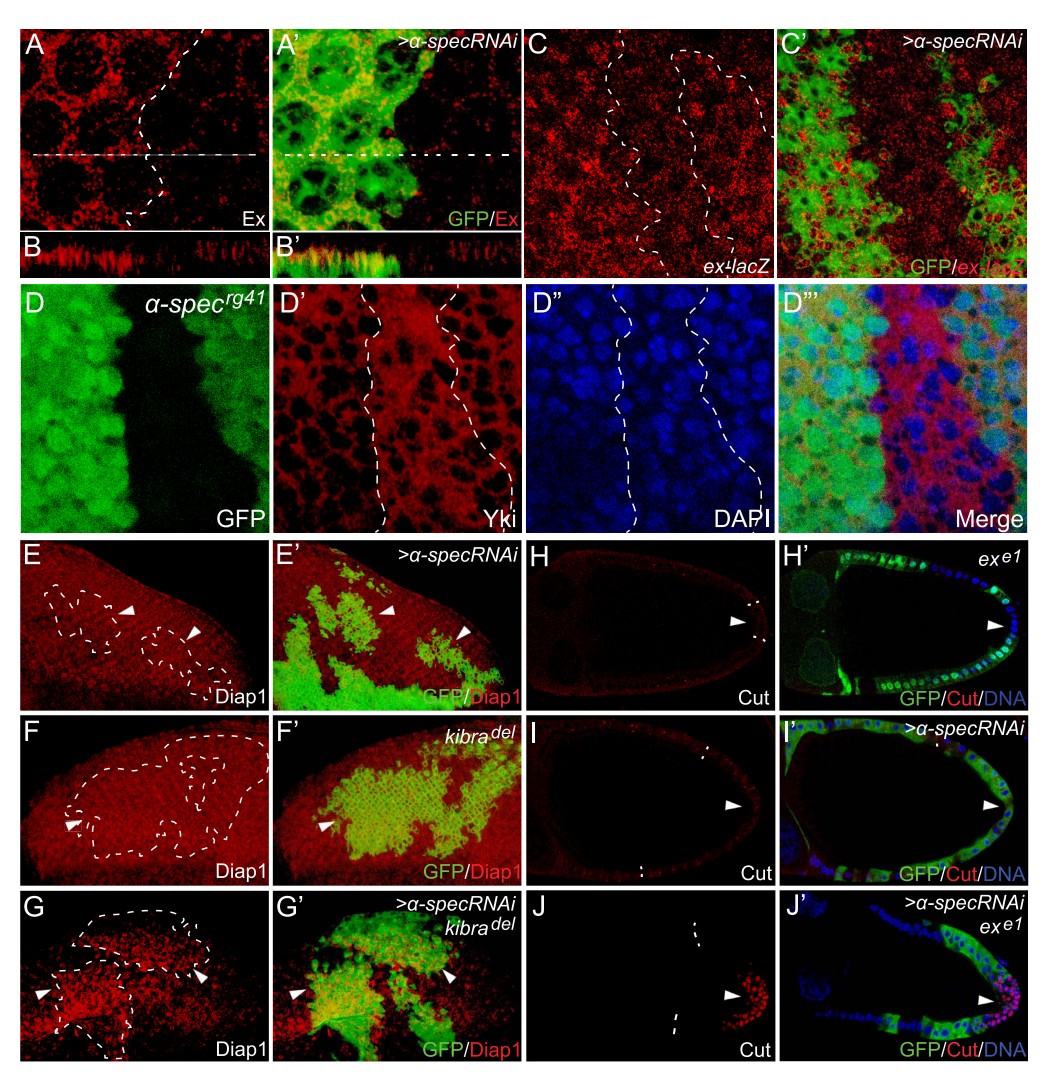

**Figure 2**. α-Spec regulates the expression of Hippo target genes. (**A–C′**) Pupal eye discs containing GFP-positive MARCM clones with *α-spec* RNAi were stained for Ex (red, in **A–B′**) or *ex-lacZ* (red, in **C–C′**). **B–B′** shows a vertical section through the eye disc in **A–A′**, in which the position of the vertical section is indicated by a straight dotted line. Note the elevated Ex level (**A** and **B**) or LacZ level (**C**) in clones with *α-spec* RNAi. (**D–D′′′**) A third instar wing disc containing GFP-negative *α-spec^rg41* mutant clones was stained for Yki protein. Note the increased nuclear Yki signal in many *α-spec^rg41* mutant cells. (**E–G′**) Third instar eye discs containing GFP-positive MARCM clones of the indicated genotypes were stained for Diap1 expression. Note the normal expression of Diap1 in clones with *α-spec* RNAi (**E–E′**) or *kibra^del* mutant (**F–F′**), and the elevated Diap1 levels in *kibra^del* mutant clones with *α-spec* RNAi (**G–G′**). (**H–J′**) Stage 10 egg chambers containing GFP-negative *ex^e1* mutant clones (**H–H′**), GFP-positive MARCM clones with *α-spec* RNAi (**I–I′**) or GFP-positive *ex^e1* mutant clones with *α-spec* RNAi (**J–J′**) were stained for Cut expression. Cut expression and multilayering of follicle cells were observed only in the *ex^e1* mutant clones with *α-spec* RNAi (**J–J′**).

The following figure supplements are available for figure 2:

**Figure supplement 1**. α-Spec regulates the expression of Hippo target genes.

**Figure supplement 2**. Loss of α-Spec does not affect the subcellular localization of Mer or Kibra.

*cut* and formation of multilayer PFCs in stage 10 egg chambers (*Meignin et al., 2007*; *Polesello and Tapon, 2007*; *Yu et al., 2008*). In contrast, inactivation of upstream regulators such as Ex leads to a milder phenotype characterized by transiently prolonged *cut* expression that disappears by stage 10 (*Yu et al., 2008*) (*Figure 2H–H'*). Similar to the *ex* mutation, *α-spec* RNAi in PFCs did not cause visible upregulation of *cut* expression in stage 10 egg chambers (*Figure 2I–I'*). However, *α-spec* RNAi in *ex* mutant clones produced multilayer of PFCs with strong *cut* expression persisted in stage 10 (*Figure 2J–J'*). Thus, spectrin functions as a widespread regulator of Hippo signaling in multiple tissue contexts. The requirement of spectrin in both imaginal discs and PFCs distinguishes it from Ft-Ds, which is required in imaginal discs but dispensable in the PFCs (*Bennett and Harvey, 2006*; *Cho et al., 2006*; *Silva et al., 2006*; *Willecke et al., 2006*; *Meignin et al., 2007*; *Polesello and Tapon, 2007*; *Yu et al., 2008*).

Next, we investigated how spectrin regulates Hippo signaling. Since spectrin usually functions as a scaffold protein at the membrane–cytoskeleton interface, we first examined the subcellular localization of three membrane-associated upstream regulators of Hippo signaling: Ex, Mer, and Kibra. Of note, similar to spectrin, these proteins regulate Hippo signaling in both imaginal discs and PFCs (*Hamaratoglu et al., 2006*; *Meignin et al., 2007*; *Polesello and Tapon, 2007*; *Yu et al., 2008*; *Baumgartner et al., 2010*; *Genevet et al., 2010*; *Yu et al., 2010*). We detected no visible changes in the subcellular localization of these proteins (*Figure 2A–B'* and *Figure 2—figure supplement 2*). Together with the synergist effect of spectrin knockdown with *ex*, *mer*, or *kibra* mutations described above, these results suggest that spectrin likely functions in parallel with Ex, Mer, and Kibra to regulate Hippo signaling.

Given recent studies implicating the actomyosin cytoskeleton as a regulator of Hippo signaling, we examined the possibility that the SBMS may regulate Hippo signaling through the actomyosin cytoskeleton, whose major components are non-muscle myosin II and F-actin. As the major force generator in most cell types, the activity of non-muscle myosin II is regulated by phosphorylation of the regulatory light chain of myosin II (MLC) by multiple kinases including Rho-associated protein kinase (Rok) (*Vicente-Manzanares et al., 2009*). Phosphorylation of MLC greatly increases the $Mg^{2+}$-ATPase activity of myosin in the presence of actin and leads to the generation of contractile forces or tension (*Somlyo and Somlyo, 2003*). Interestingly, a significant increase in p-MLC was observed in the eye imaginal disc upon RNAi knockdown of *α-spec*, *β-spec*, or *βH-spec* (*Figure 3*). Careful examination of the p-MLC signal revealed a polarized effect of spectrin knockdown that is consistent with the subcellular distribution of the different spectrin subunits (*Lee et al., 1997*) (see also *Figure 1—figure supplement 1*): reducing α-Spec led to elevated p-MLC level in both apical and basolateral cortices of the cells (*Figure 3A–B'*), but reducing βH-Spec caused increased p-MLC only in the apical cortices (*Figure 3C–D'*) while reducing β-Spec caused increased p-MLC only in the basolateral cortices (*Figure 3E–F'*). A similar increase of p-MLC was observed in the wing imaginal disc upon *α-spec* knockdown (*Figure 3I–I'*) and confirmed in mutant clones carrying the *α-spec^{rg41}* allele (*Figure 3G–G'*). Despite the increase in p-MLC level, the expression of MLC was unaffected in *α-spec^{rg41}* mutant clones, as measured by GFP staining from a genomic rescue *spaghetti squash* (*sqh*: encoding MLC in *Drosophila*)-GFP construct (*Royou et al., 2004*) (*Figure 3H–H'*), suggesting that the increased p-MLC level is not simply due to an overall increase in total MLC levels. Despite the changes of p-MLC level, phalloidin staining did not reveal gross abnormality in the overall level and integrity of actin cytoskeleton in *α-spec^{rg41}* mutant cells (*Figure 3J–J'*). Of note, the regulation of p-MLC is specific to spectrin, as changes of p-MLC were not observed in mutant clones for other Hippo pathway regulators such as *ex*, *mer*, *kibra*, and *crb*, or mutant clones for the core kinase component *wts* (*Figure 4A–E'*). These findings implicate spectrin as the only tumor suppressor identified to date that regulates Hippo signaling by modulating cortical actomyosin contractility.

To examine whether MLC activation contributes to the overgrowth phenotype produced by spectrin knockdown, we dampened MLC activation by expressing a *rok* RNAi transgene in mutant clones with *α-spec* RNAi. Indeed, loss of Rok suppressed the clonal overgrowth of *α-spec* mutant tissues (*Figure 5A–D*). Concurrent with the suppression of tissue overgrowth, loss of Rok also reversed the increased p-MLC level and the extra interommatidial cells seen in the *α-spec* mutant tissues (*Figure 5E–F'''*). In fact, the *α-spec rok* double RNAi clones resemble *rok* RNAi clones as both showed decreased growth, decreased levels of p-MLC and decreased number of interommatidial cells as compared to wild-type clones (*Figure 5F–G'''*). As an alternative to *rok* RNAi, we dampened MLC activation by expressing a dominant negative form of Rok (Rok^{KG}) (*Winter et al., 2001*) and observed

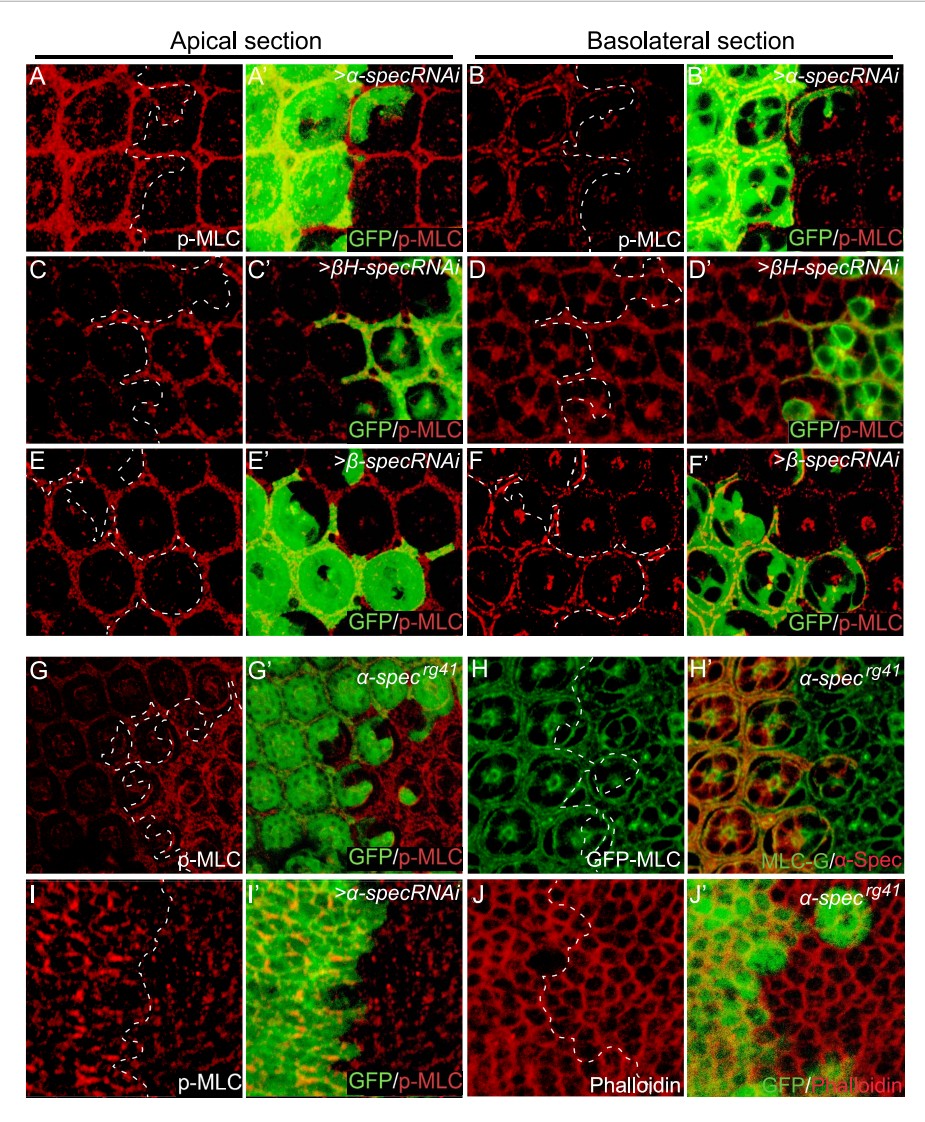

**Figure 3**. Loss of spectrin promotes the phosphorylation and activation of MLC. (**A–F'**) Pupal eye discs containing GFP-positive MARCM clones with *α-spec*, *βH-spec*, or *β-spec* RNAi were stained for phospho-MLC (p-MLC, red). For each imaginal disc, both apical section (**A–A'**, **C–C'** and **E–E'**) and basolateral confocal section (**B–B'**, **D–D'** and **F–F'**) were shown. Note the increase of p-MLC in both apical and basolateral sections of the *α-spec* mutant cells (**A–A'** and **B–B'**), the increase of p-MLC only in the apical section of *βH-spec* mutant cells (**C–C'** and **D–D'**), and the increase of p-MLC only in the basolateral section of the *β-spec* mutant cells (**E–E'** and **F–F'**). (**G–G'**) A pupal eye disc containing GFP-negative *α-spec^{rg41}* mutant clones stained for p-MLC. Note significant increase of p-MLC level in the mutant clone. (**H–H'**) A pupal eye disc containing GFP-negative *α-spec^{rg41}* mutant clones was stained for α-Spec (red) and Sqh-GFP (green). Note the similar levels of Sqh-GFP expression inside and outside the *α-spec^{rg41}* mutant clones. (**I–I'**) A Third instar wing disc containing GFP-positive MARCM clones with *α-spec* RNAi was stained for p-MLC. Note significant increase of p-MLC level in clones with *α-spec* RNAi. (**J–J'**) A Third instar wing disc containing GFP-negative *α-spec^{rg41}* mutant clones was stained for F-actin using phalloidin (red). Note the similar actin cytoskeleton organization in *α-spec^{rg41}* mutant clones.

a similar suppression of *α-spec* mutant phenotypes (data not shown). These data are consistent with spectrin acting upstream of MLC to regulate Hippo signaling. In agreement with this genetic epistasis, overexpression of Wts in *α-spec* mutant clones completely reversed the increased number of interommatidial cells without affecting the elevated p-MLC level in these clones (*Figure 4F–G'''*). These data further support our model implicating spectrin as an upstream regulator of Hippo signaling.

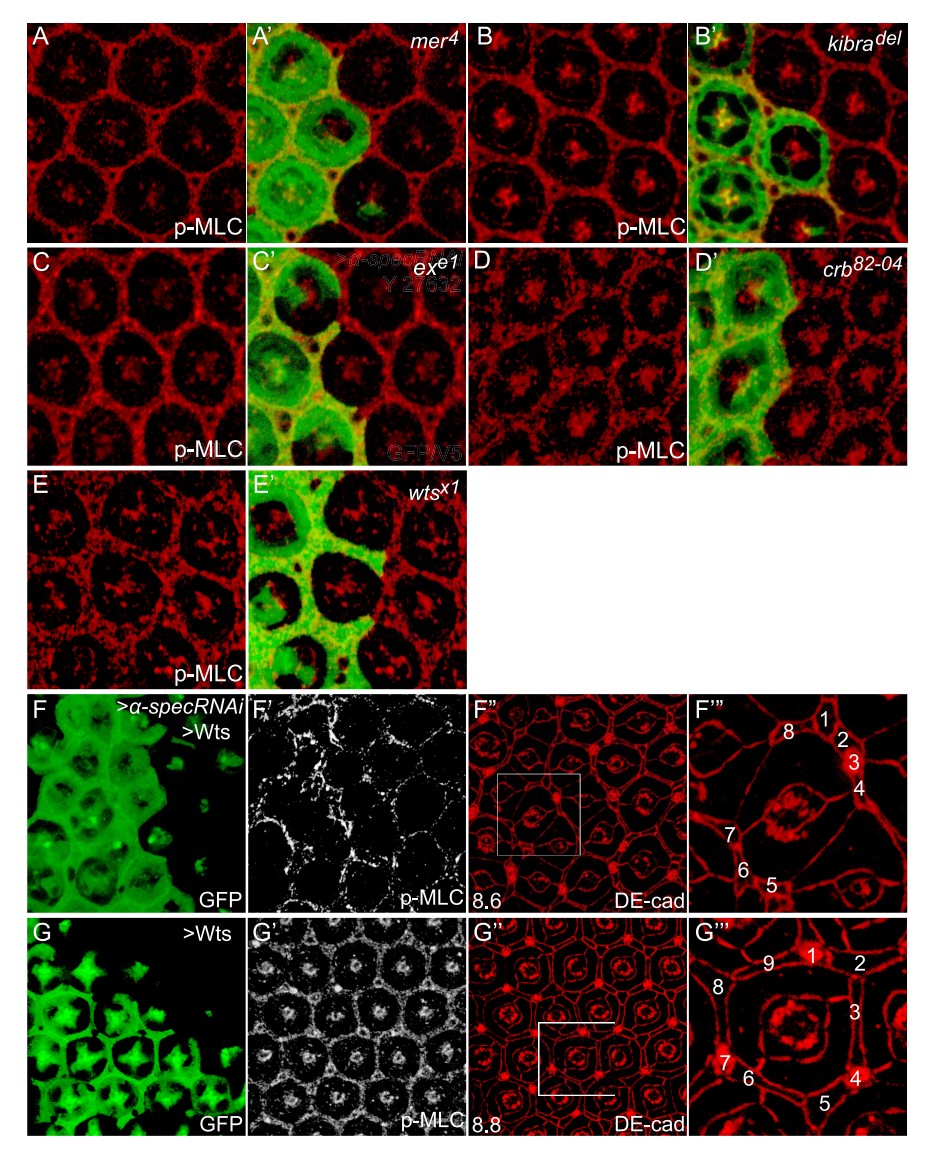

**Figure 4**. Canonical upstream tumor suppressors of the Hippo pathway do not regulate p-MLC activity. (**A**–**E'**) Pupal eye discs containing GFP-positive MARCM clones of the indicated mutations were stained for p-MLC. Note the similar levels of p-MLC inside and outside the mutant clones. (**F**–**F'''**) A pupal eye disc containing GFP-positive MARCM clones with *α-spec* RNAi and Wts overexpression, showing upregulation of p-MLC (**F'**) and decreased number of interommatidial cells (**F''**; the number in the lower left represents the average number of interommatidial cells surrounding each unit eye calculated from 20 mutant ommatidia). The magnified view of a representative ommatidium in **F''** (boxed area) is shown in **F'''**, with all the interommatidial cells marked by different numbers. Wild-type eyes have an average of 12 interommatidial cells surrounding each unit eye (*Carthew, 2007*) (see also *Figure 1E*). (**G**–**G'''**) Similar to **F**–**F'''** except that MARCM clones with Wts overexpression were analyzed. The magnified view of a representative ommatidium in **G''** (boxed area) is shown in **G'''**. Note the similar level of p-MLC (**G'**) in the clones compared to the neighboring wild-type tissues. Also note the decreased interommatidial cell number compared to wild-type eyes.

It was recently reported that increased cytoskeleton tension suppresses Hippo signaling by recruiting the Ajuba (Jub)-Wts complex to the apical junctions (*Rauskolb et al., 2014*). To investigate whether spectrin regulates Hippo signaling through the Jub–Wts complex, we examine the subcellular localization of Jub and Wts in spectrin-defective cells using Wts-V5, Wts-GFP and Jub-GFP reporters (*Rauskolb et al., 2014*). Despite the increased actomyosin activity in *α-spec* knockdown cells

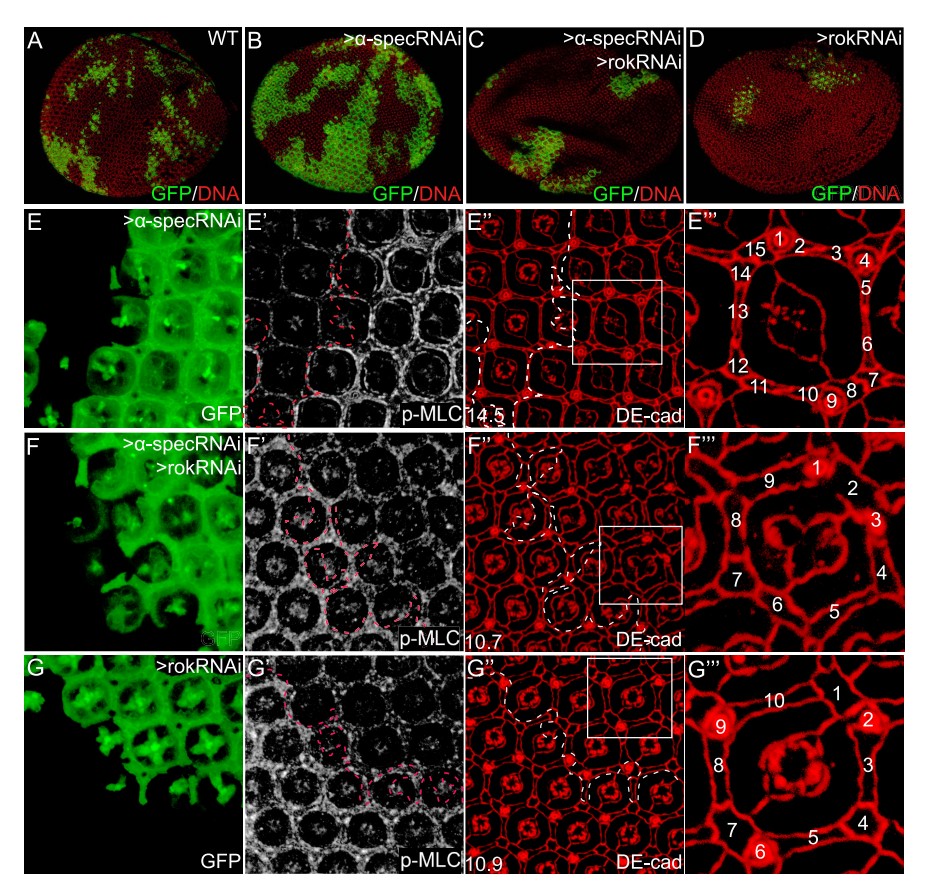

**Figure 5**. Inhibition of MLC activation suppresses the *α-spec*-deficient phenotypes. (**A–D**) Pupal eye discs containing GFP-positive MARCM clones of the indicated genotypes. Note the increased clone size resulting from *α-spec* RNAi (compare the relative representation of GFP-positive tissues in the whole eye between **B** and **A**). Also note the decreased representation of the GFP-positive clones resulting from *rok* RNAi (**D**) or *α-spec rok* RNAi (**C**). (**E–E‴**) A pupal eye disc containing GFP-positive MARCM clones with *α-spec* RNAi, showing upregulation of p-MLC (**E′**) and increased number of interommatidial cells (**E″**; the number in the lower left represents the average number of interommatidial cells surrounding each unit eye calculated from 20 mutant ommatidia). The magnified view of a representative ommatidium in **E″** (boxed area) is shown in **E‴**, with all the interommatidial cells marked by different numbers. (**F–F‴**) Similar to **E–E‴** except that MARCM clones with *α-spec* and *rok* double RNAi were analyzed. Note the decrease of p-MLC (**F′**) and interommatidial cell number (**F″–F‴**) in the clones compared to the neighboring wild-type tissues. Wild-type eyes have an average of 12 interommatidial cells surrounding each unit eye (*Carthew, 2007*) (see also *Figure 1E*). (**G–G‴**) Similar to **E–E‴** except that MARCM clones with *rok* RNAi were analyzed. Note the decrease of p-MLC (**G′**) and interommatidial cell number (**G″–G‴**) in the clones compared to the neighboring wild-type tissues.

The following figure supplement is available for figure 5:

**Figure supplement 1**. Loss of α-Spec does not affect the subcellular localization of Jub or Wts.

---

(*Figure 3I–I′*), we did not observe detectable changes in subcellular localization of Wts or Jub proteins (*Figure 5—figure supplement 1*). These findings suggest that spectrin-regulated cortical actomyosin activity may regulate Hippo signaling through a different mechanism.

The above results suggest that the *α-spec* mutant phenotypes are due to high actomyosin contractility in these cells. A prediction of this model is that elevating actomyosin activity may be sufficient to recapitulate these phenotypes. To test this hypothesis, we increased actomyosin activity directly by expressing an activated MLC (Sqh$^{EE}$), which mimics phosphorylated Sqh at T20 and S21 (*Winter et al., 2001*). The number of ECPC of ommatidium was used as a characteristic and

semi-quantitative readout of defective Hippo signaling in the pupal retina (*Figure 6*). Consistent with the hypothesis, expression of Sqh$^{EE}$ produced 3.6 ECPC, similar to the *mer* mutants (*Figure 6A–D*). Moreover, expression of Sqh$^{EE}$ in *mer* or *ex* mutant clones led to a dramatic increase in the number of ECPC (*Figure 6E–F*), suggesting a synergistic effect between cytoskeletal tension and the canonical upstream regulators of the Hippo pathway in tissue growth. This synergism was further confirmed by evaluating the expression of the Hippo target gene Diap1. While neither Sqh$^{EE}$-overexpressing clones nor *mer*$^4$ mutant clones in the pupal retina showed upregulation of Diap1 expression, *mer*$^4$ mutant clones with Sqh$^{EE}$ overexpression showed a significant elevation of Diap1 level (*Figure 6G–I''*). These results further support the view that the spectrin and myosin II-mediated cytoskeletal tension functions in parallel with the canonical upstream regulators of the Hippo pathway to regulate downstream signaling.

Finally, we examined whether the spectrin may play a conserved role in regulating Hippo signaling in mammalian cells. The α spectrin subunit is encoded by two genes in humans: *SPTA1* which is expressed mainly in erythrocytes, and *SPTAN1* which is expressed in all non-erythrocyte cells (*Moon and McMahon, 1990*; *Cianci et al., 1999*; *Berghs et al., 2000*). When the expression of *SPTAN1* was knocked down through small interfering RNA (siRNA) in MCF10A cells (a human mammary epithelial cell line), we observed increased nuclear localization of YAP compared to control cells in confluent cultures (*Figure 7A–B'''*), as well as increased cortical p-MLC level (*Figure 7C–D'*). Consistent with immunostaining, *SPTAN1* RNAi led to decreased YAP phosphorylation at the Hippo-responsive Ser 127 and Ser 381 sites compared to the control cells, and also increased p-MLC (*Figure 7E*). These results suggest that the SBMS may play a conserved role in regulating actomyosin activity and Hippo signaling in mammalian cells.

## Discussion

Although the actomyosin-mediated cytoskeletal tension has been implicated as a regulator of Hippo signaling from insects to mammalian cells, how the actomyosin cytoskeleton itself is modulated remains largely unknown. Through the identification of all the spectrin subunits as negative growth regulators (tumor suppressors) and the characterization of their mechanism of action in *Drosophila*, we have uncovered the SBMS, a cytoskeleton directly underneath the plasma membrane, as an essential regulator of the actomyosin cytoskeleton in Hippo signaling. The negative regulation of myosin II activity is unique to the spectrin, since it is not shared by other known upstream tumor suppressors of the Hippo pathway. Rather, the spectrin-Myo II pathway functions in parallel with the other upstream regulators to modulate Hippo signaling in vivo. Although spectrin was reported to regulate the Hippo pathway during the preparation and review of this paper (*Fletcher et al., 2015*; *Wong et al., 2015*), our study is the first to uncover a functional link between the SBMC and Myo II activity, or to establish the parallel relationship between spectrin-MyoII-mediated actomyosin activity and the other known upstream regulators of Hippo signaling. To our knowledge, this is the first demonstration that a structural membrane skeleton protein regulates Hippo signaling by modulating actomyosin contractility.

Our study also provides new insight into the nature of force production that is relevant to Hippo signaling. Previous studies implicating cytoskeletal tension as a regulator of Hippo signaling were based on whole-cell perturbation of the actomyosin cytoskeleton (*Dupont et al., 2011*; *Wada et al., 2011*; *Aragona et al., 2013*; *Rauskolb et al., 2014*). Since the actomyosin cytoskeleton is distributed in diverse subcellular compartments such as the cell cortex, the cytoplasm and the cell–cell junction, it was unclear from these experiments which subcellular pool(s) of the actomyosin cytoskeleton are relevant to Hippo signaling. Our observation that loss of the SBMS leads to defective Hippo signaling with elevated p-MLC levels only around the cell cortex suggests that the actomyosin cytoskeleton directly underneath the plasma membrane is functionally linked to the Hippo pathway. Interestingly, ablation of apical spectrin (βH-spec) and basolateral spectrin (β-spec) leads to localized activation of p-MLC in the apical and basolateral cortex, respectively. Yet, both ablations result in similar Hippo-related growth defects. Based on these findings, we infer that the actomyosin cytoskeleton in both membrane domains is relevant to Hippo signaling, in contrast to a previous report suggesting that only apical actin polymerization affects Hippo signaling (*Fernandez et al., 2011*).

While our study has uncovered a functional link between spectrin and actomyosin activity, the molecular mechanisms by which actomyosin activity regulates Hippo signaling remain to be determined. At present, there are contrasting views on whether cytoskeleton-mediated regulation of Yki/YAP activity

**Figure 6**. Myosin II-regulated cytoskeletal tension functions in parallel with the canonical upstream tumor suppressors to regulate Hippo signaling. (**A**–**F**) The synergistic effect of myosin II activation and loss of Ex or Mer on interommatidial cell number. Pupal eye discs of the indicated genotype were stained for DE-cad. 20 ommatidial clusters of each genotype were used for counting interommatidial cells, and the number on the lower right of each panel indicates the average number of ECPC. Note the dramatic increase of ECPC in **E** and **F**. (**G**–**I**″) The synergistic effect of myosin II activation and loss of Mer on Hippo target gene expression. Pupal eye discs containing

*Figure 6. continued on next page*

*Figure 6. Continued*

GFP-positive MARCM clones of the indicated genotypes were stained for Diap1 expression. Note the normal expression of Diap1 in *mer⁴* mutant clones (**G'**) or Sqh$^{EE}$-overexpressing clones (**H'**), and the elevated Diap1 levels in *mer⁴* mutant clones with Sqh$^{EE}$ overexpression (**I'**).

is Wts/Lats-dependent or Wts/Lats-independent (reviewed in *Low et al., 2014*). Our characterization of spectrin is more consistent with a Wts/LATS-dependent mechanism, since YAP phosphorylation is decreased by spectrin knockdown in mammalian cells (*Figure 7E*) and that overexpression of Wts can rescue the ECPC phenotype of spectrin RNAi without rescuing the high p-MLC levels in *Drosophila* (*Figure 4F*). Nevertheless, our data cannot exclude the possibility of Wts/Lats-independent mechanisms.

Previous studies have focused on the different functions of the SBMS and the actomyosin cytoskeleton in cell biology. Much of the studies on SBMS have emphasized its role in the

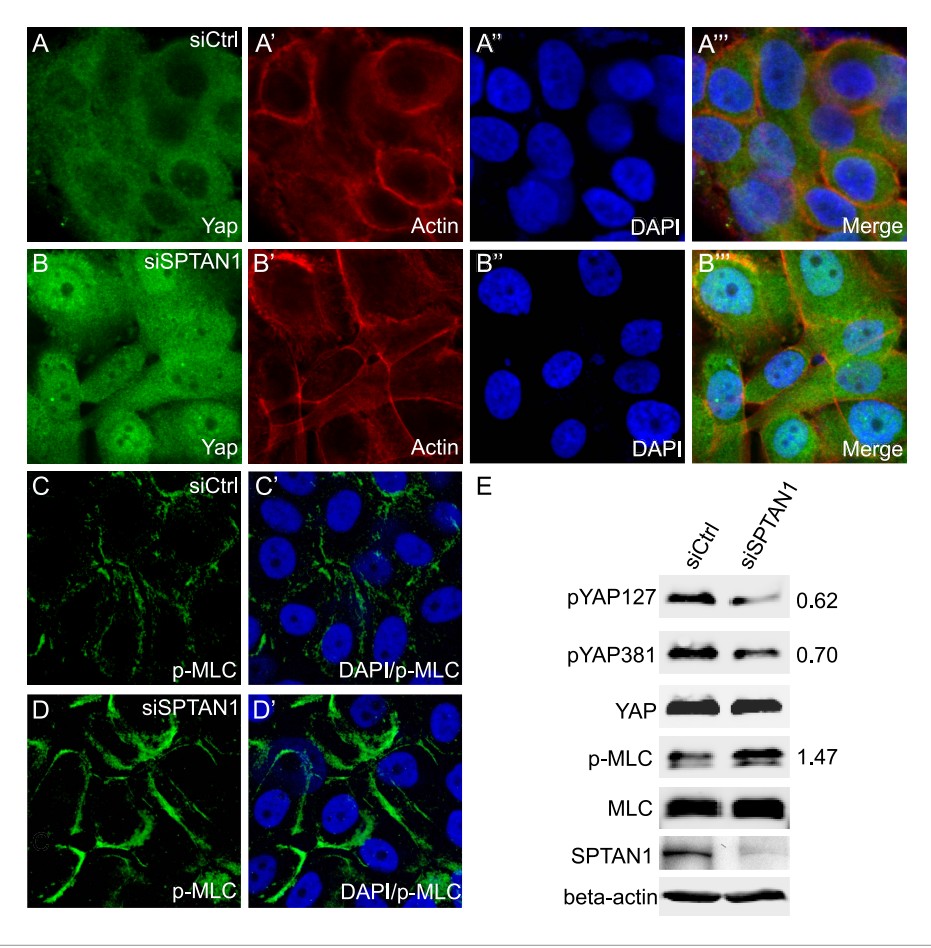

**Figure 7**. Loss of SPTAN1 results in decreased YAP phosphorylation, increased YAP nuclear localization, and increased cortical p-MLC level in MCF10A cells. (**A–B'''**) Confluent cultures of MCF10A cells treated with control RNAi or SPTAN1 RNAi were stained for YAP (green), actin (red) and the nuclear dye DAPI (blue). Note the increased nuclear YAP signal in cells with SPTAN1 RNAi. (**C–D'**) Confluent cultures of MCF10A cells treated with control RNAi or SPTAN1 RNAi were stained for p-MLC (green) and DAPI (blue). Note the increased cortical p-MLC signal in cells with SPTAN1 RNAi. (**E**) Western blot analysis of cells from **A–D'**. Quantification of p-YAP to total YAP ratio (or p-MLC to total MLC ratio) in the SPTAN1 RNAi cells, normalized to that in the control RNAi cells, is shown to the right. Note the decreased YAP S127 and S381 phosphorylation and increased p-MLC level in cells with SPTAN1 RNAi.

maintenance of plasma membrane integrity, while studies of the actomyosin cytoskeleton have implicated it in diverse cellular processes from mechanotransduction to cell migration (*Bennett and Baines, 2001*; *Lecuit and Lenne, 2007*; *Campellone and Welch, 2010*; *Lecuit et al., 2011*; *Machnicka et al., 2012*; *Takeichi, 2014*). Although spectrin subunits have been reported to physically interact with several proteins that may directly or indirectly impact the actomyosin cytoskeleton, such as F-actin, Adducin, α-Catenin and Moesin (*Gardner and Bennett, 1987*; *Pradhan et al., 2001*; *Medina et al., 2002*; *Barkalow et al., 2003*; *Xu et al., 2013*), functional coupling between these two cytoskeleton systems remains poorly understood. Our current study uncovers a previously underappreciated functional link between these two cytoskeleton systems and demonstrates that this link is physiologically important for regulating the Hippo pathway in multiple developmental contexts. We speculate that mechanical stimuli on cell membrane may be transduced first to the SBMS directly underneath the plasma membrane, which then regulates actomyosin tension to influence cell behaviors. Conversely, the actomyosin cytoskeleton may also regulate the function of the SBMS (*Fukata et al., 1999*). Understanding the detailed molecular mechanism by which the SBMS regulates Myo II activity will provide an important entry point to dissect the crosstalk between the SBMS and the actomyosin cytoskeleton in diverse cellular processes.

## Materials and methods

### Drosophila genetics

*UAS-α-specRNAi*, *UAS-β-specRNAi* and *UAS-βH-specRNAi* lines were obtained from Vienna *Drosophila* Resource Center (VDRC, stock ID 25387, 42054 and 37075). *UAS-rokRNAi* was obtained from the Bloomington *Drosophila* Stock Center (stock ID 34324). The following flies have been described previously: *α-spec*$^{rg41}$ (*Lee et al., 1993*), *β-spec*$^C$ (*Yamamoto et al., 2014*), *ex*$^{e1}$ (*Boedigheimer and Laughon, 1993*), *mer*$^4$ (*Fehon et al., 1997*), *kibra*$^{del}$ (*Yu et al., 2010*), *crb*$^{82-04}$ (*Ling et al., 2010*), *wts*$^{X1}$ (*Xu et al., 1995*), UAS-Wts (*Yin et al., 2013*), Sqh-GFP (*Royou et al., 2004*), UAS-Sqh$^{EE}$ and UAS-Rok$^{KG}$ (*Winter et al., 2001*), Wts-V5, Wts-GFP and Jub-GFP (*Rauskolb et al., 2014*). All crosses were done at 25°C. The following genotypes were used for clonal analysis:

Control MARCM clones:

UAS-GFP hs-flp; tub-Gal80 FRT40A/FRT40A; tub-Gal4/+

MARCM clones with RNAi of *α-spec, β-spec, βH-spec* or *rok*:

UAS-GFP hs-flp; tub-Gal80 FRT40A/FRT40A UAS-α-specRNAi; tub-Gal4/+
UAS-GFP hs-flp; tub-Gal80 FRT40A/FRT40A UAS-β-specRNAi; tub-Gal4/+
19A /19A tub-Gal80 hs-flp; UAS-βH-specRNAi / UAS-GFP; tub-Gal4/+
UAS-GFP hs-flp; tub-Gal80 FRT40A/FRT40A; tub-Gal4/UAS-rokRNAi

MARCM clones expressing constitutively active MLC (Sqh$^{EE}$):

UAS-GFP hs-flp; tub-Gal80 FRT40A/FRT40A UAS-sgh$^{EE}$; tub-Gal4/+

MARCM clones with *α-spec* and *rok* double RNAi:

UAS-GFP hs-flp; tub-Gal80 FRT40A/FRT40A UAS-α-specRNAi; tub-Gal4/UAS-rokRNAi

MARCM clones with both *α-spec* RNAi and Sqh$^{EE}$ overexpression:

UAS-GFP hs-flp; tub-Gal80 FRT40A/FRT40A UAS- α-specRNAi, UAS- sqh$^{EE}$; tub-Gal4/+

*mer, ex, kibra, crb* or *wts* mutant MARCM clones:

19A mer$^4$/19A tub-Gal80 hs-flp; UAS-GFP/+; tub-Gal4/+
UAS-GFP hs-flp; tub-Gal80 FRT40A/FRT40A ex$^{e1}$; tub-Gal4/+
UAS-GFP hs-flp; tub-Gal4/+; tub-Gal80 FRT82B/FRT82B kibra$^{del}$
UAS-GFP hs-flp; tub-Gal4/+; tub-Gal80 FRT82B/FRT82B crb$^{82-04}$
UAS-GFP hs-flp; tub-Gal4/+; tub-Gal80 FRT82B/FRT82B wts$^{Xl}$

*mer, ex* or *kibra* mutant MARCM clones with *α-spec* RNAi:

19A mer$^4$/19A tub-Gal80 hs-flp; UAS-GFP/UAS-α-specRNAi; tub-Gal4/+
UAS-GFP hs-flp; tub-Gal80 FRT40A/FRT40A ex$^{e1}$, UAS-α-specRNAi; tub-Gal4/+
UAS-GFP hs-flp; tub-Gal4/ UAS-α-specRNAi; tub-Gal80 FRT82B/FRT82B kibra$^{del}$

## Cell culture, siRNA transfection, and immunofluorescence staining

MCF10A cells were cultured in DMEM/F12 (Invitrogen, Carlsbad, California) supplemented with 5% horse serum, 20 ng/ml EGF, 0.5 µg/ml hydrocortisone, 10 µg/ml insulin, 100 ng/ml cholera toxin, and 50 µg/ml penicillin/streptomycin. Cells were maintained in a 37°C incubator at 5% CO$_2$. Cells were cultured on Lab-Tek II Chamber Slide (Thermo Scientific, Waltham, Massachusetts) to 80–85% confluent, and then siRNA transfections were performed twice using Lipofectamine RNAiMAX Reagent with a 24-hr interval. Cells were processed for immunoblotting or immunofluorescence 96 hr post-transfection. SMARTpool siRNA oligonucleotides toward human SPTAN1 and Non-Targeting siRNA Pool #2 (control siRNA) were purchased from GE Dharmacon (Lafayette, Colorado). For immunostaining, cells were fixed and stained following standard method using anti-YAP antibody (Novus, 1:300 dilution) and anti-p-MLC antibody (1:10, Cell Signaling Technologies, Beverly, Massachusetts). Western blotting was done using antibodies against following proteins: pYAP127 (1:1000, Cell Signaling Technologies); pYAP381 (1:1000, Cell Signaling Technologies); YAP (1:200, Sigma, St. Louis, Missouri); MLC (1:1000, Cell Signaling Technologies); SPTAN1 (1: 500, Santa Cruz Biotechnology, Dallas, Texas); and beta-actin (1:200, Sigma).

## Acknowledgements

We thank Dr Rick Fehon for Ex and Mer antibodies, Dr Nic Tapon for Kibra antibody, and the Bloomington and Vienna Stock Centers for *Drosophila* stocks. This work was supported in part by grants from the National Institutes of Health (EY015708). DP is an investigator of the Howard Hughes Medical Institute.

## Additional information

### Competing interests

DP: Reviewing editor, *eLife*. The other authors declare that no competing interests exist.

### Funding

| Funder | Grant reference | Author |
|---|---|---|
| Howard Hughes Medical Institute (HHMI) | Investigator | Duojia Pan |
| National Institutes of Health (NIH) | EY015708 | Hua Deng, Wei Wang, Jianzhong Yu, Yonggang Zheng, Yun Qing, Duojia Pan |

The funders had no role in study design, data collection and interpretation, or the decision to submit the work for publication.

### Author contributions

HD, Conception and design, Acquisition of data, Analysis and interpretation of data, Drafting or revising the article; WW, JY, YZ, YQ, Conception and design, Acquisition of data, Analysis and interpretation of data; DP, Conception and design, Analysis and interpretation of data, Drafting or revising the article

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
