## [Decision Letter]

Thank you for sending your work entitled “Spectrin regulates Hippo signaling by modulating cellular cortical tension” for consideration at *eLife*. Your article has been favorably reviewed by Janet Rossant (Senior editor and Reviewing editor) and 2 other reviewers.

The Reviewing editor and the other reviewers discussed their comments before we reached this decision, and the Reviewing editor has assembled the following comments to help you prepare a revised submission.

The reviewers agreed that this is a very interesting paper on the role of the F-actin crosslinking protein Spectrin as a regulator of fly Yki, the ortholog of mammalian YAP/TAZ. Spectrin inactivation causes wing overgrowth by modulating contractility of cortical actin thus demonstrating a linkage between spectrins, the cortical cytoskeleton and modulation of the downstream Hippo pathway. The data are clear, important and deserving publication in this journal. However there were several major concerns and some minor ones that need to be addressed before we can consider a revised manuscript.

1) All of the data presented are consistent with spectrin playing some kind of upstream role in regulating actomyosin and thus directly or indirectly regulating Hippo signaling. However, to claim, as the authors do throughout the text that “spectrin functions as a general regulator of Hippo signaling in multiple tissue contexts” is incorrect. The authors themselves clearly show in this paper that the effects of spectrin and associated cortical tension are genetically distinct from cell polarity determinants or merlin that are the more established upstream inputs of the Hippo cassette. The fact that these distinct pathways are genetically synergistic, rather than epistatic, reinforces the notion that these are parallel cues, as previously shown in mammalian cells. In fact, the conclusion that “these results further support the view that the spectrin and myosin II-mediated cytoskeletal tension functions in parallel with the canonical upstream regulators of the Hippo pathway to regulate downstream signaling” is a more correct interpretation. We would suggest that one pathway is the canonical Hippo pathway and the other is not. This is a point of novelty that should be actually highlighted.

2) There is very little in the paper to suggest that the loss of spectrin actually affects the Yki co-activator localization in the *Drosophila* system. Is there any evidence for this?

3) The *Drosophila* pupal retina has been an excellent model to study the mechanical force contribution to cell patterning during development (see references below). However, the evaluation of mechanical tension, as presented in the manuscript, is good but not robust. The major “read outs” of the tension used in this report are the p-MLC levels and the number of cells per cluster, ECPC. Indeed, the high level of p-MLC might imply high tension, but the high tension could also result in different cell shape/pattern, especially in *Drosophila* pupal retina. There is no good description and/or quantification regarding cell shapes. Moreover, there is no clear indication how the tension is related to the extra number of cells (i.e., ECPC). To improve the paper it is suggested that the retinal cell pattern should be better quantified and the pattern linked to mechanical tension, and/or to quantify the tension by laser ablation (Hayashi T, Carthew RW. Surface mechanics mediate pattern formation in the developing retina. Nature 431: 647-652, 2004. Hilgenfeldt S, Erisken S, Carthew RW. Physical modeling of cell geometric order in an epithelial tissue. Proc Natl Acad Sci USA 105: 907-911, 2008).

4) The paper demonstrates a linkage between spectrins and the role of actomyosin in regulating Hippo signaling, but it is not clear how the spectrin network is linked to the actomyosin cytoskeleton. What happens to the integrity of the cytoskeleton in spectrin mutants? What are the essential domains of spectrin required for this interaction? Also it is not clear where and how the parallel upstream Hippo pathways intersect.

5) The data on possible spectrin function in mammalian cells is very minimal: knockdown of spectrin in a mammary cell line led to increased nuclear Yap with decreased phosphorylation at Hippo-responsive sites. Do they also see effects related to cortical tension changes in mammalian cells?

---

## [Author Response]

*The reviewers agreed that this is a very interesting paper on the role of the F-actin crosslinking protein Spectrin as a regulator of fly Yki, the ortholog of mammalian YAP/TAZ. Spectrin inactivation causes wing overgrowth by modulating contractility of cortical actin thus demonstrating a linkage between spectrins, the cortical cytoskeleton and modulation of the downstream Hippo pathway. The data are clear, important and deserving publication in this journal. However there were several major concerns and some minor ones that need to be addressed before we can consider a revised manuscript*.

Thank you for the positive assessment of our manuscript and helpful suggestions. In this revision, we have conducted additional experiments in response to the reviewers’ comments. Specifically, we have added data showing: 1) nuclear accumulation of Yki in *α-spec* mutant clones in *Drosophila* wing imaginal discs (Figure 2); 2) loss of spectrin does not affect the overall integrity of cytoskeleton in imaginal discs (Figure 3); 3) loss of spectrin does not affect the localization of the Jub-Wts complex (Figure 5—figure supplement 1); 4) increased p-MLC level in mammalian cells with *SPTAN1* RNAi (Figure 7). Furthermore, in response to reviewers’ major comment #3, we have slightly modified the Title of the paper to more precisely convey our major conclusions. We hope that these additional data/changes have significantly improved our manuscript.

*1) All of the data presented are consistent with spectrin playing some kind of upstream role in regulating actomyosin and thus directly or indirectly regulating Hippo signaling. However, to claim, as the authors do throughout the text that “spectrin functions as a general regulator of Hippo signaling in multiple tissue contexts” is incorrect. The authors themselves clearly show in this paper that the effects of spectrin and associated cortical tension are genetically distinct from cell polarity determinants or merlin that are the more established upstream inputs of the Hippo cassette. The fact that these distinct pathways are genetically synergistic, rather than epistatic, reinforces the notion that these are parallel cues, as previously shown in mammalian cells. In fact, the conclusion that “these results further support the view that the spectrin and myosin II-mediated cytoskeletal tension functions in parallel with the canonical upstream regulators of the Hippo pathway to regulate downstream signaling” is a more correct interpretation. We would suggest that one pathway is the canonical Hippo pathway and the other is not. This is a point of novelty that should be actually highlighted*.

We are sorry for the confusion here. In the text “spectrin functions as a general regulator of Hippo signaling in multiple tissue contexts*”,* the word “general” was meant to emphasize that spectrin regulates the growth of multiple tissues, not just in a single tissues. In the revision, we have replaced “general” with “widespread” to avoid the confusion. As correctly summarized by the reviewers, our data support the view that spectrin and myosin II-mediated cytoskeletal tension functions in parallel with the canonical upstream regulators of the Hippo pathway to regulate downstream signaling. We highlight this point more prominently in the revision.

*2) There is very little in the paper to suggest that the loss of spectrin actually affects the Yki co-activator localization in the* Drosophila *system. Is there any evidence for this?*

Thanks for the suggestion. We have added data showing increased nuclear Yki in *α-spec* mutant clones in *Drosophila* wing imaginal discs (Figure 2).

*3) The* Drosophila *pupal retina has been an excellent model to study the mechanical force contribution to cell patterning during development (see references below). However, the evaluation of mechanical tension, as presented in the manuscript, is good but not robust. The major “read outs” of the tension used in this report are the p-MLC levels and the number of cells per cluster, ECPC. Indeed, the high level of p-MLC might imply high tension, but the high tension could also result in different cell shape/pattern, especially in* Drosophila *pupal retina. There is no good description and/or quantification regarding cell shapes. Moreover, there is no clear indication how the tension is related to the extra number of cells (i.e., ECPC). To improve the paper it is suggested that the retinal cell pattern should be better quantified and the pattern linked to mechanical tension, and/or to quantify the tension by laser ablation (Hayashi T, Carthew RW. Surface mechanics mediate pattern formation in the developing retina. Nature 431: 647-652, 2004. Hilgenfeldt S, Erisken S, Carthew RW. Physical modeling of cell geometric order in an epithelial tissue. Proc Natl Acad Sci USA 105: 907-911, 2008)*.

We agree with the reviewers’ comment that, although we observed changes in p-MLC levels, which indicate changes in actomyosin contractility, we did not measure mechanic tension directly. Measuring retinal cell shape/pattern is insufficient for this purpose since it is influenced by many factors such as cell adhesion, cell death, cell movement, not all of which are directly related to cell tension (Larson et al., Plos Computational Biology 6: e1000841, 2010). In another word, cell shape is not a more direct assay for cell tension than p-MLC. As suggested by the reviewers, the best way to quantify cell tension is by laser ablation. Although this technique has been done in fly embryos and larval discs, it is very challenging in pupal discs (we are not aware of any previous reports of this technique in the pupal eyes).

Although we did not measure mechanic tension directly by laser ablation, we did show that increased MLC activity, either due to loss of spectrin or direct overexpression of active MLC, both result in similar extra cells per cluster (ECPC) phenotype. Further, the ECPC phenotype of spectrin mutant is rescued by dampening MLC activity. Thus, although we have not measured mechanic tension directly, we can conclude that spectrin regulates Myosin II activity and this regulation underlies the role of spectrin in growth control. To more precisely convey our results, we use “actomyosin activity” or “actomyosin contractility” instead of “mechanical tension” or “cell tension” in the revised manuscript. We also modified the Title of the paper to better convey this point.

*4) The paper demonstrates a linkage between spectrins and the role of actomyosin in regulating Hippo signaling, but it is not clear how the spectrin network is linked to the actomyosin cytoskeleton. What happens to the integrity of the cytoskeleton in spectrin mutants? What are the essential domains of spectrin required for this interaction? Also it is not clear where and how the parallel upstream Hippo pathways intersect*.

How the spectrin and actomyosin networks are linked to each other is poorly understood in any epithelial system. We agree with the reviewers that this is an important question but feel that addressing this question is beyond the scope of the current paper. As suggested by the reviewers, we have examined the integrity of cytoskeleton in spectrin mutant cells and did not observe any gross abnormality (data added as Figure 3).

We agree with the reviewers that understanding how the parallel upstream Hippo pathways intersect is important. It was recently reported that cytoskeleton tension regulates Hippo signaling through the Jub-Wts complex (46). We examined this possibility but did not observe detectable changes in Jub-Wts complex in spectrin mutant cells. This data is added in the revision (Figure 5—figure supplement 1).

*5) The data on possible spectrin function in mammalian cells is very minimal: knockdown of spectrin in a mammary cell line led to increased nuclear Yap with decreased phosphorylation at Hippo-responsive sites*. *Do they also see effects related to cortical tension changes in mammalian cells?*

Thanks for the suggestion. Indeed, we also observed increased p-MLC level in mammalian cells with *SPTAN1* RNAi. We have added this data in the revision (Figure 7).